# Hitting the Sweet Spot: How Glucose Metabolism Is Orchestrated in Space and Time by Phosphofructokinase-1

**DOI:** 10.3390/cancers16010016

**Published:** 2023-12-19

**Authors:** Melissa Campos, Lauren V. Albrecht

**Affiliations:** 1Department of Developmental and Cell Biology, School of Biological Sciences, University of California, Irvine, CA 92697, USA; camposm4@uci.edu; 2Department of Pharmaceutical Sciences, School of Pharmacy & Pharmaceutical Sciences, University of California, Irvine, CA 92697, USA

**Keywords:** glycolysis, spatiotemporal, phosphofructokinase-1, subcellular, condensate, glucosome, localization, compartmentalization, cancer, optogenetics

## Abstract

**Simple Summary:**

Glucose metabolism was the first metabolic pathway to be discovered and serves a central role across all life by generating energy. Cancer cells hijack glycolysis to promote growth and metastasis. Understanding the molecular basis of glycolytic rewiring in cancers could illuminate new treatment approaches.

**Abstract:**

Glycolysis is the central metabolic pathway across all kingdoms of life. Intensive research efforts have been devoted to understanding the tightly orchestrated processes of converting glucose into energy in health and disease. Our review highlights the advances in knowledge of how metabolic and gene networks are integrated through the precise spatiotemporal compartmentalization of rate-limiting enzymes. We provide an overview of technically innovative approaches that have been applied to study phosphofructokinase-1 (PFK1), which represents the fate-determining step of oxidative glucose metabolism. Specifically, we discuss fast-acting chemical biology and optogenetic tools that have delineated new links between metabolite fluxes and transcriptional reprogramming, which operate together to enact tissue-specific processes. Finally, we discuss how recent paradigm-shifting insights into the fundamental basis of glycolytic regulatory control have shed light on the mechanisms of tumorigenesis and could provide insight into new therapeutic vulnerabilities in cancer.

## 1. Introduction

Glycolysis, originally known as the Embden–Meyerhof–Parnas pathway, is a fundamental metabolic process across all life. The net gain of glycolysis generates two ATPs and pyruvates, which subsequently leads to 36 ATPs from oxidative phosphorylation (Figure 1). Intermediates throughout the pathway also generate precursors for serine (one carbon metabolism), sugars (protein glycosylation), and nucleotide synthesis (pentose phosphate pathway) [1]. Glucose enters the cell through transporters at the plasma membrane and becomes rapidly phosphorylated to trap glucose inside [2]. There are 10 core enzymes in the glycolytic pathway that are found throughout the cytoplasm [3]. A wide range of cellular mechanisms have evolved in order to regulate glycolytic flux. For instance, glycolytic enzymes can be controlled on the level of protein expression, post-translational modification codes, and through precise spatiotemporal compartmentation [4]. The present review seeks to provide an overview of recent discoveries in this field of research. As a lens for discussing these emerging concepts, we focus on phosphofructokinase-1 (PFK1), the gatekeeper of glycolysis [5]. We highlight the emerging methodologies that have uncovered novel spatiotemporal control mechanisms such as optogenetics, a genetic approach that uses light to noninvasively control proteins rapidly and reversibly with precise spatial resolution [6,7,8]. Moreover, we discuss the implications of PFK1 that becomes dysregulated in cancer and how this new breadth of knowledge could offer vulnerabilities to be exploited for therapeutics.

## 2. PFK Protein Structure and Composition

### 2.1. PFK1 Isoforms

PFK1 is localized in the cytosol and consists of three domains with approximately 780 amino acids (85 kDa) (Figure 2). PFK1 catalyzes an irreversible reaction that commits glucose to glycolytic breakdown by converting fructose-6-phosphate (F6P) into fructose-1,6-bisphosphate (FBP) [2]. Vertebrates have three isoforms of PFK1 that share a 95% sequence homology and were named based on the tissue of their discovery, including PFK-M (muscle), PFK-L (liver), and PFK-P (platelet) [9]. Most human tissues have all three isoforms, albeit with different levels of expression. PFK-P is the most highly expressed form across all tissue types [10]. Distinct enzymatic regulation has been reported for each PFK1 isoform, despite their similar amino acid composition.

### 2.2. PFK1 Expression, Activity and Genetic Mutations

Metabolic reprogramming is a classic hallmark of cancer [11,12]. Most notably, cancerous cells exhibit an increased rate of glycolysis independent of the amount of oxygen present. This phenomenon, known as aerobic glycolysis or the Warburg effect, is thought to create a metabolic environment that supports high levels of cell proliferation through the accumulation of nutrients and free energy [2]. It is not surprising that PFK1 expression is also increased in many cancers, seeing as it is one of the key enzymes that determines glycolytic rate. It has been previously shown that increasing PFK1 expression promotes aerobic glycolysis, directly connecting PFK1 expression with the metabolic phenotype seen in cancer [13]. PFK1 expression increases in breast cancer, lung cancer, brain cancer, bladder cancer, and colon cancer [14,15,16,17,18]. This increase in PFK1 expression is not observed across all three isoforms equally, with L or P being the predominant isoforms in cancer [10]. The reason for this biased isoform expression in cancer remains unknown. Overexpression of TAp73 protein, which is an isoform of the tumor suppressor p73, is often overexpressed in tumors and increases the expression of PFK-L to promote glycolysis [13]. Additionally, the activation of the hypoxia-inducible factor pathway, which commonly promotes tumor growth, has been tied to increased PFK1 activity through allosteric activation, typically by fructose-2,6-bisphosphate (F-2,6-BP) [19,20,21,22]. 

Genetic mutations that alter glycolytic activity have been identified in PFK-P and PFK-L in several types of cancers [5,23]. The expression of PFK-P cancer mutants N426S and D564N in rat-derived breast cancer cells demonstrate an increase in PFK1 activity and lactic acid production, which is suggestive of aerobic glycolysis (Table 1) [23]. PFK-L cancer mutation in aspartate (D553N) correlates with decreased glycolysis in breast cancer, suggesting an intentional metabolic redirection to the pentose phosphate pathway [5]. Additionally, a K727A mutation of PFK-L led to constitutive activity that increased glycolysis and invasiveness in ovarian cancer (Table 1) [24]. In contrast, genetic mutations in PFK-M were reported in the 1960s to cause glycogen storage disease type VII, known as Tarui disease. Tarui disease is a rare autosomal recessive disorder that is characterized by PFK-M deficiency. A lack of the PFK1 enzyme results in an inability to break down glucose and leads to skeletal muscle deterioration and indirect kidney damage [25]. Together, the altered functions of glucose metabolism that result from genetic mutations in PFK1 highlight the fundamental importance of maintaining a tight regulation of PFK1 activities.

### 2.3. PFKFB Isoforms 

6-phosphofructo-2-kinase/fructose-2,6-bisphosphatase (PFKFB) plays an essential role in coordinating glycolytic and gluconeogenic fluctuations in metabolism [26]. PFKFB was identified after the initial discovery of F-2,6-BP as an allosteric regulator of PFK1 [26,27]. There are four isoforms of PFKFB (PFKFB1, PFKFB2, PFKFB3, and PFKFB4) that serve as bifunctional homodimers responsible for catalyzing the creation and degradation of F-2,6-BP (Figure 1). PFKFB proteins have two domains that act as independent enzymes: the N terminal domain serves as a kinase, known as phosphofructokinase-2 (PFK2), while the C terminal domain serves as a phosphatase named fructo-2,6-bisphosphatase (FBPase2) [26,28]. The two differing domain activities are regulated primarily by phosphorylation, where the addition of a phosphate simultaneously inhibits kinase activity and promotes phosphatase activity, while dephosphorylation has the opposite effect [29]. PFKFB isoforms have some degree of tissue specificity: PFKFB1 is highly expressed in skeletal and cardiac muscle as well as the liver while PFKFB2 is primarily found in cardiac muscle. PFKFB3 is expressed in most organs, while PFKFB4 is mainly expressed in the testes [30,31,32]. PFKFB isoforms only have partial tissue specificity and can still be found to be co-expressed in various tissues [32]. PFKFB isoforms are also differentiated by their kinase to phosphatase activity ratios, where certain isoforms more often catalyze one reaction over the other [26,33].

### 2.4. PFKFB Expression and Activity 

PFKFB is an essential determinant in the regulation of carbohydrate metabolism by controlling the concentration of F-2,6-BP in cells, which is the most potent activator of PFK1 activity. Therefore, its altered expression in certain cancers can lead to aberrant glucose metabolism. Certain cancers exhibit a biased isoform expression of PFKFB similarly to PFK1. PFKFB1 has not been reported to have obvious anomalous expression in cancer [30,34,35]. Meanwhile, PFKFB2 has been shown to be overexpressed in pancreas, lung, and prostate cancer [36,37,38]. PFKFB2 has also been reported to have an opposite expression pattern in colorectal cancer, where decreased PFKFB2 was correlated with poor prognosis in patients [39]. PFKFB3 and PFKFB4 are the most overexpressed and active isoforms in a plethora of cancers; these two isoforms are responsible for the common upregulation of F-2,6-BP seen in cancer cells [31,33,35,40,41,42]. PFKFB3 has the highest kinase to bisphosphatase activity ratio among all the isoforms, producing the most F-2,6-BP and triggering increased glycolytic flux [31,43,44]. Alternatively, PFKFB4 has been suggested to indirectly shunt glucose into the pentose phosphate pathway, which provides the cell with a means to combat reactive oxygen species (ROS) [33,45]. The biased expression of both PFKFB3 and PFKFB4 could allow for metabolic fine tuning that accommodates rapid growth during tumorigenesis.

## 3. PFK1 Regulation 

### 3.1. Metabolites

PFK1 plays a unique role in glycolysis on multiple levels. First, the conversion of F6P to FBP represents the slowest enzymatic step in the glycolytic pathway. On the atomic level, PFK1 has two states of the quaternary structure known as the low-activity T state and the high-activity R state [23]. An inactive dimeric PFK1 has also been reported [46]. PFK1 is allosterically regulated by metabolites [9]. ATP, citrate, and phosphoenolpyruvate (PEP) stabilize the T state to decrease catalytic activity while AMP and FBP stabilize the R state to increase PFK1 activity (Figure 2). On a cellular level, this tuning of PFK1 operates as a way to modulate glycolytic flow based on the metabolic demands of a specific cell state. For instance, when energy levels are sufficient, high levels of ATP, citrate, and PEP are produced from the TCA cycle and glycolysis, respectively. Once energy levels drop, increased levels of AMP trigger PFK1 to become more active so as to provide glucose and energy from glycolysis [47]. In the early 1980s, F-2,6-BP was identified as a novel allosteric regulator of PFK1 [27]. Now known as PFK1′s most potent activator, F-2,6-BP increases PFK1 activity while relieving ATP inhibition, which increases glucose uptake and metabolic flux [48,49]. It is interesting to note that the muscle isoform, PFK-M, has the highest binding affinities for both F6P (K_0.5_^F6P^ 147 µM) and ATP (K_0.5_^ATP^ 152 µM) while PFK-P has the lowest binding affinities (K_0.5_^ATP^ 276 µM and K_0.5_^F6P^ 1333 µM). In contrast, PFK-L demonstrates high ATP binding affinity (K_0.5_^ATP^ 160 µM) but low F6P binding affinity (K_0.5_^F6P^ 1360 µM) [9,50]. These in vitro biochemical measurements highlight the distinct properties of each PFK1 isoform and suggest that the expression of each PFK1 paralog may contribute to fulfilling the unique metabolic demands across differing cell types and tissues.

PFK1 expression has been tied to cancer in numerous ways, the most obvious being to promote glycolysis and cancerous cell growth. One possibility for this discrepancy in isoform expression may be due to isoforms’ reactivity to allosteric regulation. Each isoform has different affinities for distinct allosteric regulators, with PFK-L and PFK-P being more responsive to allosteric activators than PFK-M [9]. It has also been suggested that PFK-M possesses a stable state of activity in order to maintain basal metabolic control compared with the other two isoforms. Another possibility may be due to differences in spatiotemporal regulation where isoform region-specific clustering into cellular domains could also serve to increase glycolysis. Considering that each isoform has different sensitivity levels to activity triggers, cancer cells may be preferentially expressing isoforms that respond the most efficiently to unique metabolic cues within the tumor microenvironment.

### 3.2. Post-Translational Modifications 

Post-translational modifications (PTMs) serve to both regulate PFK1 activity and dictate localization in the cell, simultaneously in some cases. In 1975, it was first reported that PFK1 is phosphorylated in liver extracts [51]. Since this discovery, phosphorylation has been shown to promote the localization of PFK1 onto actin structures, which in turn increases enzyme activity [52]. Additionally, Akt phosphorylation at residue S386 has been shown to stabilize protein levels by blocking subsequent ubiquitination and proteasomal degradation [53]. PFK1 acetylation also drives the formation of multienzyme complexes that contribute to glucose metabolism (Table 2) [54].

Several PFK1 PTMs that occur in physiologic conditions are hijacked to promote cancer cell metabolism. In ovarian cancer, S-nitrosylation has also been reported to promote PFK1 activity by creating resistance to negative feedback regulators [55]. Hypoxia leads to PFK1 serine 529 modification with O-GlcNAcylation, which inhibits enzymatic activity and redirects glucose flux into the pentose phosphate pathway to promote growth in multiple solid tumor cancer cell lines, including breast, prostate, liver, colon, and cervical cells (Table 2) [56]. Blocking the O-GlcNAcylation of PFK1 reduced cancer cell proliferation in vitro and decreased tumorigenesis in vivo [56]. In these cases, cancer cell growth was enabled by PFK1 inhibition by shunting metabolites into the pentose phosphate pathway to generate nucleic acids for rapid proliferation while also contributing to the prevention of oxidative stress in cancer microenvironments [57,58] (Figure 1).

A biased expression of PFK1 may allow the cell to have better control over PFK1 activity and cellular growth [4]. A recent report by Rossi et al. highlighted that interactions between PFK-P and phosphoglycerate dehydrogenase (PHGDH) were critical for regulating the delicate balance between glycolysis and the sialic acid pathway. By inhibiting PFK1 activity, its substrate F6P was more readily available to be shunted into the hexosamine pathway, which allowed for cellular glycosylation that aided in metastasis. Once the cancerous cells mobilized, PFK1 activity resumed to provide nutrients and energy to allow the cell to grow. By controlling PFK1 activity, cancer cells can better adapt to both mobilization and growth in the body, two aspects that are crucial to the spread of cancer.

## 4. PFK1 Spatiotemporal Regulation

The biochemical regulation of glycolytic enzymes has been the focus of intensive research investigations. Precise biochemical reactions of PFK1 are determined by subcellular localization in the cytosol, along the plasma membrane, and even at the mitochondrial interface [59]. Localized expression of PFK1 and FBP has also been seen during certain stages of embryogenesis. In murine embryos, localization during chorioallantoic branching was correlated with a decrease in glycolysis while localization during organogenesis was tied to an increase in glycolysis at the site of higher expression [60,61,62]. This research suggests that spatiotemporal regulation is a requirement for precise metabolic control during crucial moments of growth and maturation. Localized expression of glycolytic enzymes has also been seen in other species during development, such as in avian and amphibian embryos [63,64]. Recent research has also begun to show that subcellular localization of both PFK1 and PFKFB in cancer can serve as a predictor for cancer recurrence [65]. These emerging studies highlight the need for understanding spatiotemporal regulation and its significance in development and disease. Such an architectural remodeling of PFK1 throughout the cell is rapid, dynamic, and highly reversible, which suggests that such mechanisms may have evolved as additional approaches to tune the production of metabolites depending on the tissue-specific demands.

### 4.1. Plasma Membrane

The plasma membrane is a hotspot of metabolic activity being the interface between intracellular and extracellular matrices. Glucose enters the cell at the plasma membrane through glucose transporters [66]. As the entry point for glucose into the cell, activated PFK1 at the plasma membrane ensures efficient and rapid energy production. Caveolae are plasma membrane invaginations that are driven by the activity of caveolin proteins [67]. Additionally, caveolin can also serve scaffolding functions for metabolic pathways such as glycolysis. Indeed, PFK-M is known to associate with multiple isoforms of caveolin, and PFK-M recruitment to caveolae depends on the specific Cav-3 isoform expression level. It has been reported that human mutations in Cav-3 reduce PFK-M expression and membrane recruitment in skeletal muscle tissues [68,69]. High extracellular glucose concentrations also promote the recruitment and targeting of PFK-M into caveolin-enriched caveolae domains, which suggests that PFK-M is recruited to promote glycolysis at the plasma membrane surface [69]. In line with this, PFK1 activators stabilize Cav-2/PFK-M complexes [70].

### 4.2. Cytoskeleton 

Cytoskeletal dynamics enable cellular adhesion, migration, proliferation, and homeostatic maintenance. Beyond these classic roles, cytoskeletal remodeling also coordinates cellular metabolism and metabolic enzyme activities. Actin filaments (F-actin) are linear polymers of globular actin [71]. In the case of PFK1, it has been reported for over 30 years that PFK-M interacts with actin filaments to increase PFK1 enzymatic activities [72]. This interaction is amplified by PFK1 phosphorylation, which promotes its affinity for actin [52]. Additionally, actin binds at the adenosine triphosphate (ATP) activation site on PFK1, where it acts as an allosteric activity upregulator [73]. Global glycolytic stimulators such as insulin have also been shown to drive PFK-M associations with actin [74]. Recently, the PFK-P isoform has also been reported to associate with F-actin [75]. Decreases in the product of PFK1, FBP, coupled with the dissociation of PFK-P from actin further suggests that PFK-P interactions with actin increase activity, as similarly seen with PFK-M [76]. More specifically, F-actin bundling and colocalization with PFK-P have been correlated with increased glycolysis, which indicates that PFK-P may also sense and respond to cellular mechanical cues from the cell environment [77]. In contrast, PFK1 localization with the microtubule component tubulin was found to decrease enzyme activity [78]. This is due to the preferential binding of PFK1 dimers, which are an inactive form of PFK1 [79]. Similar to F-actin, PFK1 is associated with microtubule bundling and has even been shown to drive microtubule bundling through the formation of cross-bridges [79,80,81]. While both F-actin and microtubules are essential cytoskeletal constituents of cellular architectures, these studies further highlight key functions in metabolic regulation through PFK1 and glycolysis.

### 4.3. Cytosolic Phase Condensates 

Phase condensates are membrane-less structural assemblies of proteins that form upon changes in solubility states. A recent report systematically interrogating the localization of PFK1 in vivo was performed using a *Caenorhabditis elegans* model [82]. Dynamic remodeling of PFK1 occurred in response to hypoxia. Diffuse PFK1 localized within the cytosol during normoxic conditions and was rapidly redirected into phase-separated condensates during hypoxic energy stress conditions. The authors propose that the liquid-like properties and spheroid shapes of phase condensates enable rapid internal molecular reorganization to promote PFK1 signaling dynamics. PFK1 condensates were formed through a heterologous self-association cryptochrome 2 domain and also recruited the downstream glycolytic enzyme, aldolase. Notably, PFK1 condensates were not correlated with stress granules. PFK1 in *Caenorhabditis elegans* genetically maps to the conserved region of the F674 residue, which is critical for tetramer formation and PFK multivalent interactions [82]. Together, this study in a living organism showcases how the glycolytic PFK1 enzyme can dynamically form condensates in vivo. Whether PFK1 metabolic regulation via sub-compartments occurs across mammalian systems has yet to be examined.

### 4.4. Cytosolic Filaments 

PFK-L polymerizes into filaments throughout the cytoplasm [83]. Transmission negative-stain electron microscopy revealed that filaments were an average of six tetramers long (~lengths of 65.4 nm), where tetramers are composed of two structurally distinct dimers. The unusual helical symmetry of PFK-L conformations is critical to enable the last subunit to be able to bind another subunit in a linear manner, or alternatively, to introduce a kink into the filament. These dynamic structural conformations occurred between the F6P binding pocket and the C-terminus. A chimera of the C-terminal regulatory domain of PFK-L fused to the catalytic region of PFK-P led to key insights into the isoform-selective properties driving filament formation. Indeed, this chimera delineated that the PFK-L C-terminus was responsible for determining filament assembly as it was sufficient to drive filament formation. The remaining questions involve how the enzymatic activity of PFK-L is changed upon entry into filaments and how PFK-L removal from PFK-P and PFK-M cytosolic complexes impacts glycolysis. Given that both F6P and citrate promote PFK-L filament assembly, it is tempting to speculate that this dynamic remodeling process may regulate glycolytic flux. On a larger scale, authors note that filament localization appeared as punctate in cells with sizes that could indicate a potential phase transition of PFK-L into P granules, stress granules, or aggresomes. Further investigations into this question could provide essential molecular insights into a major question in metabolism regarding how the precise location of enzyme activities directly impacts the levels of downstream metabolites.

### 4.5. Glucosome 

The term metabolon was first termed by Paul Srere as a supramolecular complex of sequential metabolic enzymes and cellular structural elements [84]. A type of metabolon, the glucosome, is defined as a multienzyme assembly that is formed by metabolic enzymes during glucose metabolism. Glucosomes are spatiotemporally controlled subcellular domains that regulate glucose flux in the cell where both glycolytic and gluconeogenic enzymes have been found inside. The importance of this structure is further highlighted by the presence of specific enzymes that catalyze rate-limiting reactions [85,86]. PFK-L has been previously reported to be a driver of glucosome formation, specifically through acetyl modifications on lysine residues [54]. PFK-L-induced glucosomes also include downstream rate-limiting enzymes such as PKM2, FBPase, and PEPCK1 (Figure 3).

The glucosome is multi-enzymatic and regulates enzymatic activities of multiple rate-limiting metabolic enzymes. While the exact function of the glucosome is currently under investigation, it has been suggested that assembly serves to quickly sequester enzymes from the cytosol and halt glucose metabolism. Given this, glucosome formation could downregulate metabolic pathways as membrane-less assemblies allow for quicker and more transient protein sequestration. Furthermore, the absence of membranes suggest that glucosome formation could provide a cost-effective mechanism to regulate cellular contents. Finally, compartmentalization of PFK1 along with other glycolytic enzymes into glucosomes could maximize metabolic rate, as having these enzymes in proximity shuttles the product of one reaction directly into the next reaction without the interference of product diffusion in the cytosol. Increasing protein proximity has been shown to increase positive cooperative effects [87]. While it was recently discovered that glucosomes are regulated by cell cycle-associated signaling pathways, this regulation is size-specific and only applies to smaller-sized glucosomes [88,89]. Thus, the full extent of underlying mechanisms that control the formation and dissociation of such glycolytic clusters still remains unknown [90].

Although the glucosome remains to be a relatively new concept in the global understanding of cellular metabolic regulation, the discovery of this signaling hub provides insights into how rate-limiting enzymes are finely tuned on a rapid time scale. Future work aimed at untangling this regulatory pathway across cell types and tissues is warranted. Filling these gaps in knowledge could have broad implications for understanding cancer cell metabolism and for designing interventions that target the required mechanisms that execute these precise metabolic shifts in glycolysis. 

### 4.6. Mechanosensation

Recent work underscored a fundamental mechanosensation mechanism through PFK1 and the regulation of downstream glycolytic activity. This key discovery was uncovered by the simple observation that F6P levels were elevated on soft substrate and coincided with reduced levels of downstream metabolites in the glycolytic pathway [77]. This led authors to explore whether protein levels of glycolytic enzymes contribute to such metabolite fluxes. Indeed, PFK1 levels were specifically reduced in conditions when F6P became elevated, while all other glycolytic enzymes remained unchanged. Importantly, authors further demonstrated that stiff substrates increased PFK1 interactions with actin bundles that segregated PFK1 from degradation enzyme TRIM21 and from the proteasomal ubiquitin machinery. This further highlights the importance of protein degradation pathways in the tuning metabolism, where previous studies have largely focused on protein expression. Furthermore, downregulation of several E3 ubiquitin ligases such as TRIM21 have been reported in cancer and could highlight how the loss of such proteins could offer an additional approach through which cancer increases glycolysis. While this report focused on human lung tumors, the stiffness of substrates represents a broad phenomenon across multiple types of cancers and could represent a more global view on mechanisms influencing glycolysis during tumorigenesis. 

## 5. Future Perspectives 

Metabolism changes as a cancer develops from a small, premalignant lesion to an aggressive primary tumor and then metastasizes. Overall, PFK1 activity is closely tied to cancer, particularly cancers of the female reproductive system [15,24,55,91,92,93,94,95,96]. Additionally, the downregulation of PFK1 in cancer has also shown to be advantageous for creating more metabolite availability for other biosynthetic pathways [56,97]. Knowledge of how PFK1 is fundamentally regulated and how this regulation goes awry in cancers will provide insight into understanding context-dependent metabolic fluidity [12,98]. Both the up- and downregulation of PFK1 can promote cancer cell growth; insights on these alternate pathways and appreciating their context-dependent use will elucidate novel mechanisms and targets for cancer treatment.

For instance, optogenetics provide exquisite spatial resolution and fast reversibility to acutely and dynamically dissect a metabolic pathway action [6,7,8]. Fluorescence single-cell microscopy has also been emphasized as a classic and indispensable tool capable of monitoring subcellular protein localization with great target specificity [99]. Overall, the enzymatic activity of PFK1 is tightly orchestrated across cell types and tissues through multiple spatiotemporal mechanisms. While many spatiotemporally controlled processes are dedicated to tuning PFK1 activity, how these regulatory principles contribute to metabolic stages and tumor types are still beginning to unfold. PFK1 is an irreversible, essential component of glycolysis, which suggests that interventions aimed at PFK1 could offer an important vulnerability for disrupting glycolytic-driven cancer cell metabolism. Furthermore, all three mammalian PFK1 isoforms generate FBP; however, the unique highly specific regulation of each isoform suggests that the moonlighting functions of PFK1 may also contribute to processes in cancer that have yet to be uncovered. As organisms became more complex during evolution, cells may have devised subcellular localization as a dynamic mechanism to provide tighter regulation on cellular processes. Thus, the cell may have evolved such approaches in order to compensate for the irreversibility of PFK1 enzymatic activity. 

## 6. Conclusions

In conclusion, this review highlights the numerous ways that PFK1 is spatiotemporally regulated. We draw attention to the importance of this regulation and isoform discrimination for possible therapeutic targets in cancer. We also elucidate new methodologies for further studying and understanding PFK1 compartmentalization. Both PFK1 and PFKFB are tightly regulated in cells to control glycolytic flux, and aberrant enzyme regulation in cancer leads to the dysregulation of glycolysis. Both increases and decreases in glucose metabolism can promote cancer cell growth and proliferation, depending on cell context. Spatiotemporal regulation is also emerging as an important type of enzyme regulation that can have profound effects on cell metabolism and cell growth. While the connection between spatiotemporal regulation and aberrant metabolic regulation in cancer is not clear, investigating the relationship between the two topics can uncover novel regulatory mechanisms in cancer metabolism that can provide new therapeutic targets for clinical treatment. 

## Figures and Tables

**Figure 1 cancers-16-00016-f001:**
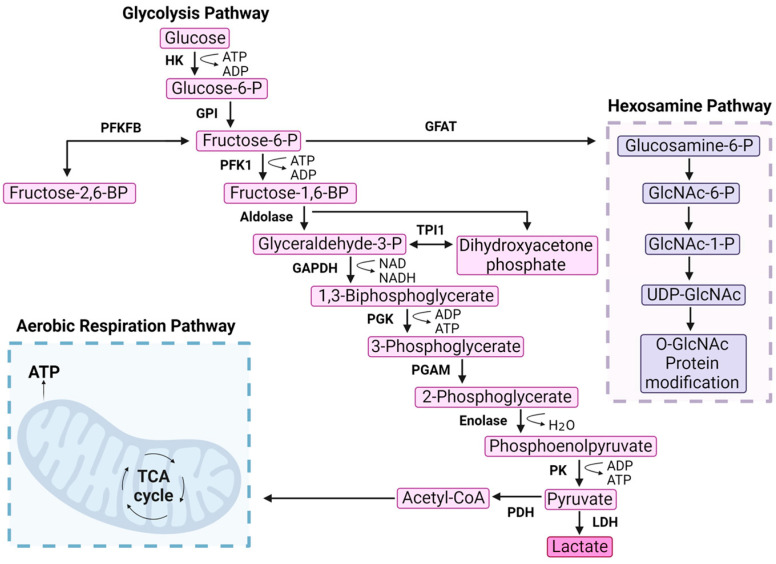
Phosphofructokinase-1 is the gatekeeper of glycolysis. Glycolysis is a multistep pathway that converts glucose into usable energy for the cell. Glucose enters the cell through transporters and is phosphorylated by HK. The first committed irreversible step of glycolysis is catalyzed by PFK1 that phosphorylates fructose-6-phosphate (F6P) to generate fructose-1,6-bisphosphate (FBP). Several side products are produced from intermediates throughout the pathway, as highlighted by F6P entry into the hexosamine pathway via glutamine fructose-6-phosphate amidotransferase (GFAT). F6P can also be converted into fructose-2,6-bisphosphate. Aerobic respiration is included on the bottom left as the entry point of acetyl-CoA. Hexokinase (HK), phosphoglucoisomerase (GPI), 6-phosphofructo-2-kinase/fructose-2,6-bisphosphatase (PFKFB), phosphofructokinase-1 (PFK1), aldolase, glyceraldehyde-3-phosphate dehydrogenase (GAPDH), phosphoglycerate kinase (PGK), phosphoglycerate mutase (PGAM), enolase, pyruvate kinase (PK), lactate dehydrogenase (LDH), pyruvate dehydrogenase (PDH), glutamine fructose-6-phosphate amidotransferase (GFAT), triosephosphate isomerase 1 (TPI1).

**Figure 2 cancers-16-00016-f002:**
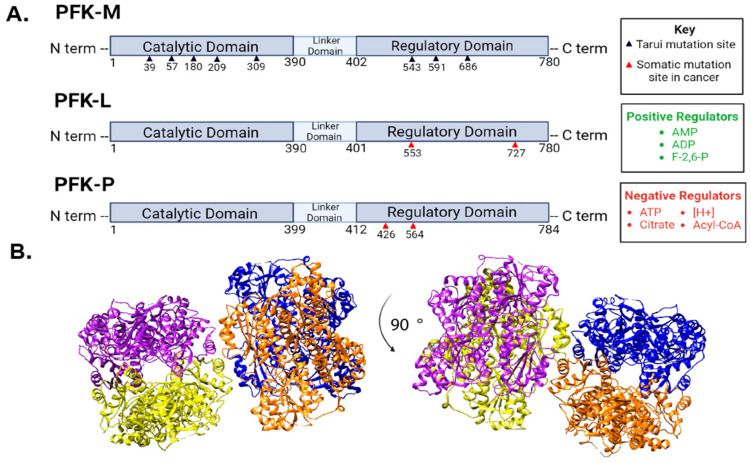
Phosphofructokinase-1 isoforms and regulation. (**A**) There are three PFK1 isoforms that each have three distinct domains. PFK-M and PFK-L are 780 amino acids long, while PFK-P is 784 amino acids long. PFK-M is the only isoform to contain mutation sites involved in Tarui disease, denoted by black triangles below PFK-M. Red arrows denote reported cancer-related mutations in PFK-L and PFK-P discussed in the text. All three isoforms respond to the listed regulators, although some isoforms are more sensitive to certain regulators than others. [H+] indicates an increase in protons or acidity as a PFK1 negative regulator. Fructose-2,6-bisphosphate (F-2,6-BP). (**B**) Representative images of PFK1 structure from different angles (90-degree rotation) taken from protein database entry 4XYK with subunits color-coded using UCSF Chimera Software. Each of the four PFK1subunits are depicted in a different color (blue, orange, yellow, and purple).

**Figure 3 cancers-16-00016-f003:**
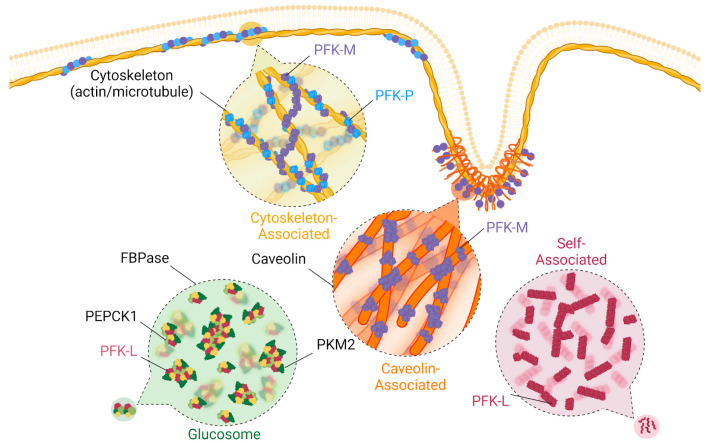
Spatiotemporal regulation of phosphofructokinase-1. PFK1 can be spatiotemporally regulated in a variety of ways, which seem to vary based on isoform type. Isoform PFK-L can self-associate into filaments in the cytosol in response to both positive and negative regulators, as well as drive glucosome formation with a variety of other metabolic enzymes. PFK-M and PFK-P are known to associate with actin, with PFK-M creating bridges between different actin filaments and between microtubules. PFK-M can associate with caveolin as well.

**Table 1 cancers-16-00016-t001:** Phosphofructokinase-1 mutations in cancer impact enzymatic activity. The mutations displayed are not comprehensive but reflect those discussed in the text.

PFK	Mutation	Residue Location	Mutation Effect on PFK1	References
PFK-L	D→N K→A	553 727	Decrease Activity Increase Activity	[23]
PFK-P	N→S D→N	426 564	Increase Activity Increase Activity	[5,24]

**Table 2 cancers-16-00016-t002:** Summary of phosphofructokinase-1 post-translational modifications (PTM) and their effect on enzyme activity.

Isoform	PTM Type	Residue Location	Modification Effect on PFK1	References
PFK-P	Phosphorylation	S386	Increase activity	[53]
PFK-L	Acetylation	K689	Formation of glycolytic clusters	[54]
PFK-M	S-nitrosylation	Cys351	Increase activity	[55]
PFK-M PFK-L PFK-P	O-GlcNAcylation	S530 S529 S540	Inhibit activity	[9,56]

## Data Availability

Not applicable.

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
