# Peer review of "Hitting the Sweet Spot: How Glucose Metabolism Is Orchestrated in Space and Time by Phosphofructokinase-1"

_cancers, 2023, doi:10.3390/cancers16010016_

Round 1
Reviewer 1 Report
Comments and Suggestions for Authors
The authors have written a short review in a crisp style focusing on phosphofructokinase-1 (PFK1), a key enzyme in glycolysis, along with PFK1 Protein Structural details and its spatiotemporal regulation. However, few suggestions for further improvement are:
1. Briefly explain the term ‘Optogenetics’ in the introduction section.
2. Add a table enlisting the PTMs along with the type and position.
3. Few more keywords should be added.
4. A three-dimensional PFK1 protein structure highlighting the 4 subunits will add value to the manuscript.
5. Add a table discussing the cancer-associated somatic mutations and their impacts on the proteins structure and function.
6. Minor language improvement and correction of typographical improvements is advised.
Comments on the Quality of English LanguageMinor improvement suggested.
Author Response
Reviewer 1 - Overview: “The authors have written a short review in a crisp style focusing on phosphofructokinase-1 (PFK1), a key enzyme in glycolysis, along with PFK1 Protein Structural details and its spatiotemporal regulation. However, few suggestions for further improvement are:”
Rev. 1: Major Point 1: “Briefly explain the term ‘Optogenetics’ in the introduction section.”
Response to Point 1: We thank the reviewer for this suggestion and are in complete agreement. We now include the following text on pages 1-2 lines 43-46 that incorporates the definition of optogenetics into the introduction: “We highlight the emerging methodologies that have uncovered novel spatiotemporal control mechanisms such as optogenetics, a genetic approach that uses light to noninvasively control proteins rapidly and reversibly with precise spatial resolution [6–8].”
Rev. 1: Major Point 2: “Add a table enlisting the PTMs along with the type and position”.
Response to Point 2: We thank the reviewer for this wonderful suggestion. We completely agree that this would improve readership. Please see page 7 line 246 that now includes a specific table to outline each PTM type and residue location.
Rev. 1:Major Point 3: “Few more keywords should be added”
Response to Point 3: Thank you for this suggestion. We agree that including additional keywords could help increase searchability and increase readership across communities. We now include the following keywords on page 1 lines 26-27: “glycolysis; spatiotemporal; phosphofructokinase-1; subcellular; condensate; glucosome; localization; compartmentalization; cancer; optogenetics”
Rev. 1: Major Point 4: ”A three-dimensional PFK1 protein structure highlighting the 4 subunits will add value to the manuscript.”
Response to Point 4: Thank you for this suggestion. We completely agree that a three-dimensional structure would greatly enhance the review. Figure 2 on page 6 now depicts the structure of PFK-P from protein database entry 4XYK. The figure depicts the protein from two different angles, and the four subunits are multi-colored for easier distinction.
Rev. 1: Major Point 5: ”Add a table discussing the cancer-associated somatic mutations and their impacts on the proteins structure and function.”
Response to Point 5: Thank you for this suggestion, we agree this table is needed to better convey information on cancer-associated somatic mutations and their impacts on protein function. A new table has been added page 3 line 118 that includes this information.
Rev. 1: Major Point 6: ”Minor language improvement and correction of typographical improvements is advised.”
Response to Point 6: Thank you for bringing this to our attention. We have now ensured that typographical corrections are addressed throughout the manuscript.

Reviewer 2 Report
Comments and Suggestions for Authors
As the main metabolic pathway of all organisms, glycolysis plays an important role in various pathophysiological processes. This review focuses on phosphofructokinase-1 (PFK1), an important molecule involved in the fateful step of oxidative sugar metabolism, and attempts to shed light on the mechanisms of tumorigenesis. However, this paper focuses on the description of PFL1 molecules, including the molecular structure and components of the protein, regulation and spatiotemporal regulation. The content of the tumorigenic mechanism is less described, and appropriate supplements are suggested. In addition, the title of the article may not reflect the main content of this review.
Author Response
Reviewer 2 - Overview: “As the main metabolic pathway of all organisms, glycolysis plays an important role in various pathophysiological processes. This review focuses on phosphofructokinase-1 (PFK1), an important molecule involved in the fateful step of oxidative sugar metabolism, and attempts to shed light on the mechanisms of tumorigenesis.”
Reviewer 2 - Major Points:
Rev. 2: Major Point 1: “However, this paper focuses on the description of PFL1 molecules, including the molecular structure and components of the protein, regulation and spatiotemporal regulation. The content of the tumorigenic mechanism is less described, and appropriate supplements are suggested.”
Response to Point 1: This is a wonderful suggestion from the reviewer and we are in complete agreement that including additional text on the tumorigenic mechanisms could be valuable for readers. To specifically address this, we now include the following text on page 3 line 96-101, which reads: “Overexpression of the TAp73 protein, which is an isoform of tumor suppressor p73 and often overexpressed in tumors, increases the expression of PFK-L and promotes glycolysis [13]. Additionally, activation of the hypoxia-inducible factor pathway, which commonly promotes tumor growth, has been tied to increased PFK1 activity through allosteric activation, typically by F-2,6-BP [19–22] .”
We also included a section that discusses PFKFB in cancer on page 4 lines 149-163 “Therefore, its altered expression in certain cancers can lead to aberrant glucose metabolism. Certain cancers exhibit biased isoform expression of PFKFB, similar to PFK1. PFKFB1 has not been reported to have obvious anomalous expression in cancer [30,34,35]. Meanwhile, PFKFB2 has been shown to have altered expression in pancreas, lung and prostate cancer, where it was shown to be overexpressed [36–38]. PFKFB2 has also been reported to have an opposite expression pattern in colorectal cancer, where decreased PFKFB2 was correlated with poor prognosis in patients [39]. PFKFB3 and PFKFB4 are the most overexpressed and active isoforms in a plethora of cancers; these two isoforms are responsible for the common upregulation of F-2,6-P seen in cancer cells [31,33,35,40–42]. PFKFB3 has the highest kinase to bisphosphatase activity ratio among all the isoforms, producing the most F-2,6-P and triggering increased glycolytic flux [31,43,44]. Alternatively, PFKFB4 has been suggested to indirectly redirect glucose into the pentose phosphate pathway, providing the cell with means to combat reactive oxygen species (ROS) [33,45]. Biased expression of both PFKFB3 and PFKFB4 could allow metabolic fine tuning to accommodate for rapid growth during tumorigenesis”.
We have also included a table on page 3 line 118 specifying cancer mutations of PFK1 discussed in the text to further highlight PFK1 involvement in cancer.
Additionally, on page 11 lines 422-424 we added “While this report focused on human lung tumors, stiffness of substrates represents a broad phenomenon across multiple types of cancers and could represent a more global view on mechanisms influencing glycolysis during tumorigenesis.”
Rev. 2: Major Point 2: “ In addition, the title of the article may not reflect the main content of this review.”
Response to Point 2: Thank you very much for the constructive feedback. We agree that revision of the title would improve clarity for readership. We now revised our title to read “Hitting the Sweet Spot: How Glucose Metabolism is Orchestrated in Space and Time by Phosphofructokinase1”.

Reviewer 3 Report
Comments and Suggestions for Authors
This very brief review of glucose metabolism in tumour cells claims to address spatiotemporal control mechanisms. It reviews some aspects of PFK1 regulation in cancer cells.
Major criticisms
There is a fairly glaring omission in this review in the sense that, although the focus is firmly on PFK1 and indeed the contributions made by other glycolytic enzymes to the regulation of glucose metabolism are ignored, the regulation by PFK2 activity has not been addressed at all. It really is not possible to review the field of research regarding PFK1 and cancer without referring to the work done on PFK2 in this tissue type. An extra section should be added to summarise recent and historical research on PFK2 isoforms.
I find the title to be quite misleading since it seems to allude to a review that I would have expected to be far more wide-ranging than this. The title implies that regulation of glucose metabolism in cancer cells in general would be addressed whereas the manuscript deals only with some fairly narrow aspects of PFK1 regulation. The title ought to be revised to reflect more accurately the scope of the review.
Minor criticisms
There are some inaccuracies:
Lines 136-139: I am unclear on what is meant by “affinity”. Are the micromolar values given the Kms? If so, this should be clearly stated. Km is not generally considered to be a measure of affinity but reflects the stability of the ES complex when derived using the steady state assumption.
Figure 2 legend: The negative regulator of PFK1 related to lactate production is generally considered to be the proton rather than the lactate molecule itself.
Lines 127-128: It is not correct to describe the T state as “inhibited” and the R state as “activated”. The T to R transition represents a reversible conformational change which may be brought about by a range of intracellular conditions.
Line 58: The statement “distinct regulation and enzymatic regulation” is very unclear.
Lines 58 and 79 have superfluous hyphens.
Line 213: “to” is missing.
Comments on the Quality of English LanguageThis very brief review of glucose metabolism in tumour cells claims to address spatiotemporal control mechanisms. It reviews some aspects of PFK1 regulation in cancer cells.
Major criticisms
There is a fairly glaring omission in this review in the sense that, although the focus is firmly on PFK1 and indeed the contributions made by other glycolytic enzymes to the regulation of glucose metabolism are ignored, the regulation by PFK2 activity has not been addressed at all. It really is not possible to review the field of research regarding PFK1 and cancer without referring to the work done on PFK2 in this tissue type. An extra section should be added to summarise recent and historical research on PFK2 isoforms.
I find the title to be quite misleading since it seems to allude to a review that I would have expected to be far more wide-ranging than this. The title implies that regulation of glucose metabolism in cancer cells in general would be addressed whereas the manuscript deals only with some fairly narrow aspects of PFK1 regulation. The title ought to be revised to reflect more accurately the scope of the review.
Minor criticisms
There are some inaccuracies:
Lines 136-139: I am unclear on what is meant by “affinity”. Are the micromolar values given the Kms? If so, this should be clearly stated. Km is not generally considered to be a measure of affinity but reflects the stability of the ES complex when derived using the steady state assumption.
Figure 2 legend: The negative regulator of PFK1 related to lactate production is generally considered to be the proton rather than the lactate molecule itself.
Lines 127-128: It is not correct to describe the T state as “inhibited” and the R state as “activated”. The T to R transition represents a reversible conformational change which may be brought about by a range of intracellular conditions.
Line 58: The statement “distinct regulation and enzymatic regulation” is very unclear.
Lines 58 and 79 have superfluous hyphens.
Line 213: “to” is missing.
Author Response
Reviewer 3 - Overview: “This very brief review of glucose metabolism in tumour cells claims to address spatiotemporal control mechanisms. It reviews some aspects of PFK1 regulation in cancer cells.”
Reviewer 3 - Major Points:
Rev. 3: Major Point 1: “There is a fairly glaring omission in this review in the sense that, although the focus is firmly on PFK1 and indeed the contributions made by other glycolytic enzymes to the regulation of glucose metabolism are ignored, the regulation by PFK2 activity has not been addressed at all. It really is not possible to review the field of research regarding PFK1 and cancer without referring to the work done on PFK2 in this tissue type. An extra section should be added to summarise recent and historical research on PFK2 isoforms.”
Author Response to Point 1: We appreciate this feedback and deeply apologize for this oversight on our part. We agree that this omission is unacceptable and have made substantial revisions to our manuscript to correct this grave error. We now include a discussion of PFK2 on a biochemical level and in the context of cancer. These new additions can be found on page 4 lines 126-163.
On page 4 lines 126-144, we now include a summary of the critical roles of PFK2 activity in glucose metabolism. This text now reads as follows: “6-Phosphofructo-2-Kinase/Fructose-2,6-Bisphosphatase (PFKFB), plays an essential role in coordinating glycolytic and gluconeogenic fluctuations in metabolism [26]. PFKFB was identified after the initial discovery of F-2,6-BP as an allosteric regulator of PFK1 [26,27]. There are 4 isoforms of PFKFB (PFKFB1, PFKFB2, PFKFB3, PFKFB4) that are bifunctional homodimers responsible for catalyzing the creation and degradation of F-2,6-BP (Figure 1). PFKFB proteins have two domains that act as independent enzymes; the N terminus domain serves as a kinase, known as phosphofructokinase-2 (PFK2), while the C terminus domain serves as a phosphatase, named fructo-2,6-bisphosphatase (FBPase2) [26,28]. The two differing domain activities are regulated primarily by phosphorylation, where addition of a phosphate simultaneously inhibits kinase activity and promotes phosphatase activity, and dephosphorylation has the opposite effect [29]. PFKFB isoforms have some degree of tissue specificity: PFKFB1 is highly expressed in skeletal and cardiac muscle as well as the liver while PFKFB2 is primarily found in cardiac muscle. PFKFB3 is expressed in most organs while PFKFB4 is mainly expressed in the testes [30–32]. PFKFB isoforms only have partial tissue specificity and can still be found co-expressed in various tissues [32]. PFKFB isoforms are also differentiated by their kinase to phosphatase activity ratio, with certain isoforms more often catalyzing one reaction over the other [26,33].”
On page 4 lines 147-163, we now include recent reports of PFK2 activity as they relate to tumorigenesis and cancer cell metabolism. This text now reads as follows: “PFKFB is an essential determinant in the regulation of carbohydrate metabolism by controlling the concentration of F-2,6-BP in cells, which is the most potent activator of PFK1 activity (Figure 2). Therefore, its altered expression in certain cancers can lead to aberrant glucose metabolism. Certain cancers exhibit biased isoform expression of PFKFB, similar to PFK1. PFKFB1 has not been reported to have obvious anomalous expression in cancer [30,34,35]. Meanwhile, PFKFB2 has been shown to have altered expression in pancreas, lung and prostate cancer, where it was shown to be overexpressed [36–38]. PFKFB2 has also been reported to have an opposite expression pattern in colorectal cancer, where decreased PFKFB2 was correlated with poor prognosis in patients [39]. PFKFB3 and PFKFB4 are the most overexpressed and active isoforms in a plethora of cancers; these two isoforms are responsible for the common upregulation of F-2,6-P seen in cancer cells [31,33,35,40–42]. PFKFB3 has the highest kinase to bisphosphatase activity ratio among all the isoforms, producing the most F-2,6-P and triggering increased glycolytic flux [31,43,44]. Alternatively, PFKFB4 has been suggested to indirectly redirect glucose into the pentose phosphate pathway, providing the cell with means to combat reactive oxygen species (ROS) [33,45]. Biased expression of both PFKFB3 and PFKFB4 could allow metabolic fine tuning to accommodate for rapid growth during tumorigenesis.”
Additionally, we now include in PFK2 in the glycolytic pathway in figure 1 on page 2
Rev. 3: Major Point 2: “I find the title to be quite misleading since it seems to allude to a review that I would have expected to be far more wide-ranging than this. The title implies that regulation of glucose metabolism in cancer cells in general would be addressed whereas the manuscript deals only with some fairly narrow aspects of PFK1 regulation. The title ought to be revised to reflect more accurately the scope of the review.”
Author Response to Point 2: Thank you very much for bringing this to our attention. We agree that revision of the title would improve clarity for readership. We now revised our title to read “Hitting the Sweet Spot: How Glucose Metabolism is Orchestrated in Space and Time by Phosphofructokinase1”.
Reviewer 3 - Minor Points:
Rev. 3: Minor Point 1: “Lines 136-139: I am unclear on what is meant by “affinity”. Are the micromolar values given the Kms? If so, this should be clearly stated. Km is not generally considered to be a measure of affinity but reflects the stability of the ES complex when derived using the steady state assumption.”
Author Response to Minor Point 1: We thank you for bringing this to our attention and agree that the term “affinity” does not clearly convey the information in the review. We have modified this term in lines 181-184 on page 5 to now read “binding affinity” and included the symbol “K0.5” to denote the values given are equilibrium constants.
Rev. 3: Minor Point 2: “Figure 2 legend: The negative regulator of PFK1 related to lactate production is generally considered to be the proton rather than the lactate molecule itself.”
Author Response to Minor Point 2: We thank you deeply for this constructive feedback. We have updated both figure 2 and the figure 2 legend on page 6 to reflect proton concentration is what is considered the PFK1 negative regulator.
Rev. 3: Minor Point 3:“Lines 127-128: It is not correct to describe the T state as “inhibited” and the R state as “activated”. The T to R transition represents a reversible conformational change which may be brought about by a range of intracellular conditions.”
Author Response to Minor Point 3: We appreciate your feedback and have adjusted our phasing to correct our descriptions of the T and R state in lines 168-170 and 172-173 on page 5. The lines now read “On the atomic level, PFK1 has two states of quaternary structure known as the low activity T state and the high activity R state [23].” and “ATP, citrate, and phosphoenolpyruvate (PEP) stabilize the T state to decrease catalytic activity while AMP and FBP stabilize the R state to increase PFK1 activity.”
Rev. 3: Minor Point 4:“Line 58: The statement “distinct regulation and enzymatic regulation” is very unclear.”
Author Response to Minor Point 4: Thank you for your suggestion, we agree this wording is unclear and have replaced the statement with the following found in page 2 line 61-62 “Distinct enzymatic regulation has been reported for each PFK1 isoform, despite their similar amino acid composition.”
Rev. 3: Minor Point 5:“Lines 58 and 79 have superfluous hyphens.”
Author Response to Minor Point 5: We thank you for bringing this to our attention, superfluous hyphens have been removed. Lines 60-61 on page 2 and line 106-108 on page 3 now read “Most human tissues have all three isoforms, albeit with different levels of expression. PFK-P is the most highly expressed form across all tissue types [10]” and “PFK-L cancer mutation in aspartate (D553N) correlates with decreased glycolysis in breast cancer, suggesting an intentional metabolic redirection to the pentose phosphate pathway [5]. ”
Rev. 3: Minor Point 6: “Line 213: “to” is missing.”
Author Response to Minor Point 6: Thank you for bringing this to our attention, the word “to” is now been added to line 267 on page 8 to read “Such architectural remodeling of PFK1 throughout the cell is rapid, dynamic, and highly reversible, which suggests that such mechanisms may have evolved as additional approaches to tune the production of metabolites depending on the tissue-specific demands.”

Reviewer 4 Report
Comments and Suggestions for Authors
The manuscript by Campos et al. is a review article summarizing the available knowledge on phosphofructokinase-1 (PFK1) in the context of recent theories on cellular organization and regulation of cellular metabolism. In my opinion, the authors approached the topic in a novel way and handled it well. The text is clear. The titles of the chapters fully correspond to the content they contain. The literature has been well selected. I have no major comments, only a few suggestions for the authors. I miss in the paper a reference to the participation of PFK1 in the regulation of metabolic changes accompanying embryogenesis. In my opinion, it would have been worth including. Next, the authors discuss the topic of controlling glycolytic regulation but using PFK1 as an example. Perhaps this should have been included in the title of the manuscript.
Author Response
Reviewer 4 - Overview: “The manuscript by Campos et al. is a review article summarizing the available knowledge on phosphofructokinase-1 (PFK1) in the context of recent theories on cellular organization and regulation of cellular metabolism. In my opinion, the authors approached the topic in a novel way and handled it well. The text is clear. The titles of the chapters fully correspond to the content they contain. The literature has been well selected. I have no major comments, only a few suggestions for the authors.”
Rev. 4: Minor Point 1: “I miss in the paper a reference to the participation of PFK1 in the regulation of metabolic changes accompanying embryogenesis. In my opinion, it would have been worth including.”
Author Response to Minor Point 1: We deeply appreciate this suggestion and are in complete agreement. We now include new references and information detailing the key importance of this enzyme during processes of development. This new text can be found on page 7 line 255-265 and reads as: “Localized expression of PFK1 and FBP has also been seen during certain stages of embryogenesis. In murine embryos, localization during chorioallantoic branching was correlated with a decrease in glycolysis while localization during organogenesis was tied to an increase in glycolysis at the site of higher expression [60–62]. This research suggests spatiotemporal regulation is a requirement for precise metabolic control during crucial moments of growth and maturation. Localized expression of glycolytic enzymes have also been seen in other species during development, such as avian and amphibian embryos [63,64]. Recent research has also begun to show that subcellular localization of both PFK1 and PFKFB in cancer can serve as a predictor for cancer recurrence [65]. These emerging studies highlight the need for understanding spatiotemporal regulation and its significance in development and disease. ”
Rev. 4: Minor Point 2: ”Next, the authors discuss the topic of controlling glycolytic regulation but using PFK1 as an example. Perhaps this should have been included in the title of the manuscript.”
Author Response to Point 2: Thank you very much for bringing this to our attention. We agree that revision of the title would improve clarity for readership. We now revised our title to read “Hitting the Sweet Spot: How Glucose Metabolism is Orchestrated in Space and Time by Phosphofructokinase1”.

Round 2
Reviewer 3 Report
Comments and Suggestions for Authors
I find the review to be much improved. It is now both wider-ranging and more focused.
Comments on the Quality of English LanguageThere remains some use of unscientific and imprecise language and editing by a specialist is recommended.